# Promoting Health and Food Literacy through Nutrition Education at Schools: The Italian Experience with MaestraNatura Program

**DOI:** 10.3390/nu13051547

**Published:** 2021-05-04

**Authors:** Beatrice Scazzocchio, Rosaria Varì, Antonio d’Amore, Flavia Chiarotti, Sara Del Papa, Annalisa Silenzi, Annamaria Gimigliano, Claudio Giovannini, Roberta Masella

**Affiliations:** 1Center for Gender-Specific Medicine, Istituto Superiore di Sanità, Viale Regina Elena 299, 00161 Rome, Italy; rosaria.vari@iss.it (R.V.); antonio.damore@iss.it (A.d.); sara.delpapa8@gmail.com (S.D.P.); annalisa.silenzi@iss.it (A.S.); claudio.giovannini@iss.it (C.G.); 2Center for Behavioral Sciences and Mental Health, Istituto Superiore di Sanità, Viale Regina Elena 299, 00161 Rome, Italy; flavia.chiarotti@iss.it; 3AMG Edutainment, Viale Egeo 14, 00144 Rome, Italy; annamaria.gimigliano@gmail.com

**Keywords:** health literacy, education, food, students, healthy lifestyle

## Abstract

MaestraNatura is an innovative nutrition education program aimed at both enhancing awareness about the importance of a healthy food–lifestyle relationship and the ability to transfer the theoretical principles of nutrition guidelines to everyday life. The educational contents of the program resulted from the analysis of the answers to a questionnaire submitted to students aged 6–13 in order to assess their degree of knowledge about nutritional facts. Educational paths were specifically designed and implemented to address the main knowledge gaps identified through the analysis of the answers and were then tested for teachers’ satisfaction in a sample of 56 schools in the north, centre, and south of Italy, involving 790 classes, 600 teachers, and 15,800 students. The results showed an approval rating from teachers from 90% to 94%. Said paths were designed for primary (6–10 years old) and first-level secondary (11–13 years old) school students. In addition, in a pilot study carried out in nine Educational Institutes located in an area close to Rome (Lazio region), a specific path was tested for effectiveness in increasing students’ knowledge about fruit and vegetables by conducting questionnaires before (T0) and after (T1) the didactic activities. Results showed a significant increase in right answers at T1 with respect to T0 (z = 2.142, *p* = 0.032). Fisher’s exact probability test showed an answer variability depending on the issue considered. In conclusion, this work could be considered as a first necessary step toward the definition of new educational program, aimed at increasing food literacy and favouring a healthier relationship with food, applicable in a widespread and effective manner, also outside of Italy.

## 1. Introduction

Chronic non-communicable diseases (NCDs) are the leading cause of death worldwide [1,2]. The main NCDs risk factors are unhealthy lifestyles, mostly bad eating habits and physical inactivity [3,4]. For this reason, the increasing incidence of NCDs is a huge challenge for a sustainable health system that intends to make universal prevention its main tool for protecting people’s health [5]. The need for preventive interventions and policies in the nutritional field concerns all age groups, because the negative effects associated with inadequate lifestyles affect all age groups worldwide. In Italy, although overweight and obesity in children have been reported to follow a decreasing trend from 2008 to 2016 [6], they still affect 30–40% of children under 18 years of age [7]. As any other behaviour, the development of the eating behaviour starts very early in life in response to a range of personal, social, economic, and environmental factors. Once acquired, eating behaviours are very hard to change [8,9,10]. 

Health literacy (HL) is defined as “the degree to which individuals can obtain, process, and understand the basic health information and services they need to make appropriate health decisions” [11]. HL is a determinant of health as it favours the adoption of correct lifestyles, the adherence to therapies, and the appropriate access to health services [12]. Furthermore, there is a growing interest about food literacy (FL) defined as a set of skills and knowledge related to food, which enables people to make informed choices about food and nutrition for improving their own health [13]. The big challenge, therefore, is to start very early with nutrition education programs to encourage the adoption of adequate lifestyles. School appears to be the most eligible setting to implement strategies aimed at improving students’ diets and food choices that play a pivotal role in promoting health [14,15]. 

Studies carried out in the last years suggest that the most successful school-based nutrition education interventions must be intensive, long lasting and comprehensive, and take account of environmental changes at school as well as family involvement and support. Research also suggests such interventions should be theory-based and incorporated into regular school curricula and activities [16,17]. 

Despite the profusion and widespread dissemination of guidelines for a healthy nutrition both in the US and Europe, the prevalence rates of overweight and obesity have been increasing [18,19], which is evidence of the scarce influence that mere information can exert on the modification of behaviour patterns. In 2008, the European Parliament resolution on the “White Paper on nutrition, overweight and obesity related health issues” indicates a multilevel and comprehensive approach to be the best way to fight obesity among the EU population. It has been pointed out the need of European programs on research, health, education, and lifelong learning, as an important step in an overall strategy to address diet-related chronic diseases in Europe. The resolution also highlighted the importance of actions aimed at improving the HL of citizens and the need of a broader educational strategy, for example by means of lessons on diet and health in primary schools [20]. In line with this vision, since 2013, the EU has promoted a school program to favour the consumption of fruit in children by distributing fresh fruit to the primary schools accordingly with the EU ‘School fruit and vegetable scheme’. The scheme also suggests the need of educational measures, including lessons, farm visits, school gardens, tasting and cooking [21]. However, much still needs to be done to reach the full potential of food and nutrition education [22]. In particular, a new paradigm is needed that goes beyond school class-based transmission of basic, generic information about food and nutrition to promote an active, hands-on learning and skill development to deal with food and nutrition in real life settings. To the best of our knowledge, this is the first study showing a nutrition education program that can be spread and easily adapted everywhere, and that allows to standardize the intervention going beyond the traditional frontal lesson which belongs to the category of passive learning together with reading, listening, and watching movies, characterized by low percentage of knowledge retention [23]. The purpose of this study was to design an effective, innovative nutrition education program, namely MaestraNatura (MN), aimed at increasing HL, and FL in particular, among primary and first-level secondary school students (6–13 years old). Final objective having students develop a balanced relationship with food together with the ability to transfer the theoretical principles contained in dietary guidelines to the actual context of a daily diet.

## 2. Methods

### 2.1. Ethical Aspects

The data were collected according to the parental written informed consent obtained for the participation of their children, in agreement with both ethical and legal (personal data protection) requirements of the Italian law. The study was explained to the participants before the start, by meeting with the teachers and providing leaflets to the parents through the schools. 

The data collected were pseudonymised soon after the data quality control assigning a univocal numerical code to each subject in order to allow the connection of data collected on the same subject before and after the educational plan. The study was not a clinical trial nor did it gather any genetic data or biological samples. For these reasons, it was not compulsory to submit the study to the IRB review. The study was part of a larger educational initiative (innovative protocols for food education in primary and lower secondary schools) approved by the Ministry of Health, General Direction Food Safety and Nutrition, which supported the program and subsequently the study. 

### 2.2. Study Design

The methodological approach was built upon the active participation of students in experimental activities at school, and the involvement of parents—or rather the family as a whole—in practical applications (cooking). The project activities were organized in three phases. The first phase of the study was devoted to exposing major misconceptions about food by administering a simple multiple-choice questionnaire to the students aged 8–13 years (Table 1, Panel A and B). To children aged 6–7 years, only Panel B of the questionnaire was administered and the questions were asked orally by the teachers. We also interviewed teachers and parents for criticism and expectations about a nutrition education program and used the collected information to set up the didactic plans, which were then submitted to the teachers for assessment. These steps were repeated a few times for content quality improvement by modifying inadequate parts and strengthening others. This phase was implemented in the Lazio region, and involved 25 schools, 200 classes, 230 teachers, and about 4000 students (aged 6–13). During the second phase, we tested the new education program for acceptance among teachers, and extended the activities to another five regions, including schools in northern (Piemonte, Veneto), central (Lazio) and southern (Campania, Basilicata) Italy. Seven towns of varying size (Torino, Verona, Padova, Roma, Benevento, Potenza, Avigliano) were involved, for a total of 56 schools, 790 classes, 600 teachers, and 15,800 students (aged 6–13) (Figure 1). The adherence and the level of satisfaction to the activities proposed by the program, were evaluated by interviewing the teachers. To evaluate the overall judgement on the project, a 0–5 scale was used; the question score > 2 was considered as positive. The developed contents of the didactic paths, as well as additional information, experiments, and recipes, were provided through a web platform (www.maestranatura.org, accessed on 3 May 2021) that teachers, students and parents could easily access. The web platform was structured to become an actual learning management system. The choice of using an information technology (IT) tool to disseminate contents allowed for cost reduction and real-time updating, not to mention the sharing of experience.

Finally, in order to test the effectiveness of the implemented didactic paths in increasing food knowledge, a pilot study was carried out with 1235 students (aged 8–10) attending 61 primary classes (3rd, 4th, and 5th classes) at 9 Educational Institutes (EI) located in a small town close to Rome, testing the educational path “It Is Easy To Say Vegetables”. We chose these classes because the contents of the path were in line with their curricular programs. The path included two didactic power point presentations on “Seeds & Fruits” and “Food Chains”, and four experimental and practical activities aimed at increasing knowledge and familiarity with vegetables (Seed’s germination and dissemination; Draw a vegetable identity card; Create a vegetable garden from kitchen waste; Extract chlorophyll from leaves). In addition, some recipes on seasonable vegetables were provided in order to favour interactions between children and parents, and to induce the students to experiment new tastes. All the planned activities were carried out throughout the school year. To evaluate the effectiveness of the path in increasing the students’ knowledge about vegetables and fruit, they were asked to fill out the panel B of the multiple-choice questionnaire (Table 1, Panel B) at the beginning and at the end of the school year. 

### 2.3. Statistical Analysis

Categorical data are presented as absolute and percent frequencies for any question of the questionnaire, both at the beginning (time 0, T0) and at the end (time 1, T1) of the study. Percentages of correct response across the questions of the questionnaire are summarized by median, minimum and maximum, separately at T0 and T1. For any child and for any question, the responses given at T1 have also been classified as changed (from wrong, W0 to correct C1 or, conversely, from correct C0 to wrong W1) or unchanged (correct or wrong at both T0 and T1). For any question and at any time, differences between sexes or classes in the percentage of correct response were assessed by the Fisher exact probability test, because the low expected frequencies in some of the contingency tables make the chi-square test not always applicable. The same test was used for each question to assess if the percentage of “improving” children (W0 to C1) was different from the percentage of “worsening” children” (C0 to W1).

Finally, the percentages of correct answer at T0 were compared to the percentages of correct answer at T1 across the questions of the questionnaire using the Wilcoxon matched-pairs signed ranks test.

For all tests, *p* < 0.05 was considered statistically significant. The significance levels were reported both with and without Bonferroni’s correction, which was applied to take into account the increase in Type I error probability due to multiple tests, where appropriate. STATA 16.0 was used for the analyses.

## 3. Results

### 3.1. Preliminary Assessment of the Main Knowledge Gaps in Nutritional Facts

The results of the questionnaire (Table 1) highlighted relevant knowledge gaps on the origin and function of foods, metabolism, the role of water and macro/micronutrients, and the energy issues. For instance, 37% of the students (11–13 years old) correctly identified tomatoes as fruit, while only 24% and 16% of them identified pumpkin and courgette, respectively, as the fruit of the plant. Among the students of primary school classes (8–10 years old), 29% correctly identified courgettes but defined radish a fruit instead of a root. The analysis of the answers showed that, in general, the students seemed to have relied more on intuition than knowledge, giving their responses on the basis of colour and shape. For instance, the courgette, green and oblong, was very frequently classified as a stem. The situation was a little better with potatoes and carrots: 25% of the students aged 6–9 years and 40–50% of those aged 10–13 years correctly identified potatoes as tubers and carrots as roots. Regarding the origin of food, about 49% of students of the second and third classes (7–8 years old) and 40% of the students of the fourth and fifth classes (9–10 years old) believed that yoghurt does not derive from milk but rather from vegetables. More than half (60%) of the primary school students (8–10 years old) thought that milk does not contain water, protein being its main component over water and sugar. Finally, only 5% of the students (11–13 years old) correctly answered the questions on the energy, whereas most of them affirmed that steaks and vitamins (in pills) are foods that provide energy faster.

### 3.2. Definition of the Nutrition Education Contents

These findings helped us define what topics had to be discussed in detail and what basic concepts were worthy of consideration. In conclusion, at the end of the first two years of activity, we implemented a very effective, innovative nutrition education program aimed at increasing HL, and FL in particular, in primary and first level secondary school students (6–13 years old). The didactic paths specifically designed for each class of primary (6–10 years old) and first-level secondary schools (11–13 years old) are shown in Table 2. The nutrition education contents were defined according to WHO [24] and national guidelines [25] while those about sustainable diet were from FAO and United Nations [26,27]. In primary school classes, from the first to the fourth (6–9 years old), the following topics are addressed: the handling and processing of food, the discovery of water as an essential element for life, the identification of the different parts of the plant and their functions together with the knowledge about variety and seasonality of vegetables. In the fifth class, the differences between food and nutrients, and food groups are introduced in order to start learning how to combine food in a balanced daily menu. In the first class of first-level secondary school (11–12 years old), food waste, environmental footprint and sustainable diet issues are discussed in depth. Finally, in the second class of first-level secondary school (12–13 years old), the digestive process and the different organs of the human body that take part in it are discussed along with a further study of nutrients and their function in human metabolism. 

On the whole, MN sets the student as main target, the school as the ideal gateway for the educational program, the teacher as a major player to promote and support the didactic activities, and parents as indispensable actors for transferring the basic concepts of healthy diet guidelines from theory to practice.

The MN program features didactic contents that taken together represent a complete teaching plan strictly connected to the different science programs specifically designed for each class. The contents are meant to be distributed gradually along the entire scholastic path, to allow for a progressive development of the scientific issues by adapting them to the age of children.

Each educational path has the following contents: (a)Units for teachers that comprise texts and power point presentations on the themes of the course.(b)Practical activities and experiments to be carried out in the classroom with (i) illustrations and explanations facilitating the achievement of the expected results; (ii) precautions to be taken; (iii) the list of the materials and the time required to carry out the experiment.(c)Practical activities as homework with the involvement of household adults.(d)Concept maps that sum up the basic concepts covered by the didactic paths in a simplified schematic view to provide the full picture of a complex process in a simple way.(e)Questionnaires with a variable number of multiple-choice answers to evaluate the acquired knowledge.

After defining all the educational plans, we evaluated the completing rate for the proposed activities and the approval rating among the teachers involved in the study. The percentage of teachers’ positive judgements on the proposed educational contents and practical activities was significantly high (over 90%) (Table 3). The same table shows the percent of fully carried out activities. As regard the overall judgement of the projects, the 100% of teachers expressed a positive judgement. The program also received some criticism mostly regarding the management of printed contents, the composition of some of the kits to carry out the experiments, and the interaction with parents. This survey allowed us to improve the program and reorganize the distribution of contents and questionnaires, which were then transferred to the e-learning platform.

### 3.3. Pilot Study to Test One of the Educational Paths

#### 3.3.1. Assessment of Students’ Basal Knowledge on Fruit and Vegetables

A pilot study was carried out in a small town close to Rome to assess the increase in knowledge on fruit and vegetables after attending the educational path ‘It’s easy to say vegetables’. Students were given a multiple-choice questionnaire (Table 1, Panel B) at the beginning and at the end of the didactic activities. Out of the 1235 students enrolled in the study, 826 (67%) filled in the questionnaire at T0. Upon analysing the students’ answers to the questions, it became evident that the level of basal knowledge varied considerably depending on the subject of the question. The percentage of correct answers varied from a minimum of 22% (What do seeds need to develop?) to a maximum of 81% (roots allow the plant to…). There were no gender differences regarding the level of basal knowledge, other than for few topics, such as tomato and fennel, for which the percentage of boys providing the correct answers was significantly higher than for girls (data not shown). Finally, there were not differences in the level of knowledge with students’ age, unless for some topics (fennel, flower, seeds) for which the percentage of correct answers significantly increased with age (*p* < 0.001, *p* = 0,007, *p* = 0.055, respectively) (Table 4). 

#### 3.3.2. Evaluation of the Knowledge Increase at the End of Didactical Activities

Out of the 826 filling in the questionnaire at T0, 246 students (30%) answered the questionnaire at T1. At the end of the activities the total knowledge increased, with a median increase of about 12% (range from −13.8% to 32.9% for flower and tomato, respectively). The Wilcoxon test applied to the answers to all the questions showed a significant overall improvement of right answers at T1 with respect to T0 (z = 2.142, *p* = 0.032). Specifically, for seven questions out of 10 there was an increase in the right answer ranging from +7.7 to +32.9%, for two questions the percentage was quite similar at T0 and T1 (change equal to +2.4 and −2.4%), and for one out of 10 questions there was a decrease of −13.8% (Table 5). 

With respect to the assessment of the efficacy in increasing knowledge in the 246 children, Fisher’s exact probability test showed that the percentage of “improving” children ((W0 to C1)/W0) was significantly higher than the percentage of “worsening” children ((C0 to W1)/C0) for questions 1, 6, 7, and 8 (*p* < 0.001); the two percentages did not differ for questions 3, 4, and 5, while the percentage of “improving” children was significantly lower than that of “worsening” children for questions 2 (*p* < 0.020), 9 (*p* < 0.047), and 10 (*p* < 0.001) (Figure 2). 

## 4. Discussion

MN stands as an innovative nutrition education program that uses food as a didactic tool to stimulate scientific thought and the students’ awareness on the importance of healthy dietary habits. MN tries to connect different fields of knowledge, such as biology, physics, chemistry, history, ecology, environmental sciences, anthropology, and taste education, to eventually achieve adequate levels of HL. Actually, food can be considered from many different points of view: as a tool to observe natural and chemical–physical phenomena, as a main determinant of the planet survival because of its impact on the environment, as a product and expression of human culture, and as a vehicle for social relationships starting from very early life when it represents the strongest bond between a child and their parents.

MN program is a model of mixed-mode learning that appears to be especially suitable these days. In view of the information spread through old and new media and technologies, in fact, we need to balance information redundancy and the increasing speed of learning processes with the growth in learning difficulties and functional illiteracy that in Italy involves about 20% of the young population (16–24 years old) [28]. A large part of the adult population is not able to understand the complexity of reality, and similar difficulties affect children that might react with diverse cognitive disorders. The survey conducted among Italian students aged 6–13 years to define their knowledge about food and nutrition showed serious shortcomings. A certain degree of heterogeneity in the students’ responses was found depending on the issue considered and the age of participant; taken as a whole, the results of the preliminary survey were rather discouraging. Nevertheless, we started from those results to build up a comprehensive, multifaceted educational program to provide an answer to those needs by using a completely different teaching approach. Firstly, MN proposes a systemic-constructivist approach that aims to facilitate the comprehension of complexity. To face a problem from a systemic point of view is to seek connections with similar problems in different areas and, only later, to explore the distinctive characteristics of the starting problem [29]. This approach simplifies the knowledge process without trivializing it and seeks to be effective in leading to a progressive, self-generating learning typical of the constructivist approach [30]. It also favours cooperative learning, a successful teaching strategy in which small groups of students with different levels of ability engage in a variety of learning activities to improve their understanding of a subject [31]. Finally, the MN method takes into consideration Dale’s cone of learning that indicates the active learning techniques as the best base to support the acquiring of knowledge [23]. Considering purely the health aspect of nutrition education, the traditional approach embraced by most interventions is that it is enough to feed scientific information on food and nutrition to the population in order to induce a behavioural change and the adoption of a healthy lifestyle in any person. However, this approach has been questioned because of its poor effectiveness. For this reason, the American Dietetic Association stated that new nutrition education interventions must adopt methodologies able to produce effective changes in dietary habits and not to disseminate solely nutrition information [32]. Furthermore, an interesting paper reviewing the intervention programs carried out in the last years to counteract childhood obesity reports that the most effective ones were those conducted at school in the age range 6–12 and focused to modify one behaviour at a time. Didactic contents promoting healthy nutrition, physical activity, food processing, and safety, among others, appeared to be more effective when originally included in the scholastic plan and delivered in multiple sessions during the school year [33]. Finally, to create an environment facilitating behavioural changes (e.g., offering healthy food like fruit and nuts to the students) and to support parents in improving their relationship with their children also by sharing time together, e.g., cooking, are highlighted as relevant points to get good results [34]. 

The MN program takes into account these suggestions and introduces some novelties in the traditional educational approach. First of all, the educational path direction: it does not start from the food pyramid to guide behaviour towards healthy eating. On the contrary, the understanding of the food pyramid is the endpoint to be reached after gaining knowledge of nutritional facts in order to understand the guidelines for healthy eating. Focusing on health in general, MN avoids emphasis on concepts like “healthy body weight” or “good/bad” foods and promotes the whole person without neglecting children’s psychological and emotional aspects [35], which should be reconciled in integrated intervention programs aimed at the prevention of obesity and eating disorders [36]. Another distinctive aspect is that knowledge is acquired through experience in the eight-year period of primary and secondary school (6–13 years old). Finally, an ambitious objective we set is to define the real effectiveness of educational contents in increasing FL. In this regard, the pilot study carried out to evaluate the efficacy of the path ‘It Is Easy To Say Vegetables!’ in improving the knowledge on fruit and vegetables, provided interesting and encouraging results. The significant increase in right answers obtained after the path execution with respect to those obtained before, indeed, support the educational effectiveness of the path tested. However, by analysing the answer variability, the students failed more the answers to the questions about the plant functionality than those about plant recognizing. The variability found in some answers, evidenced by Fisher’s test, led us to rethink the wording of the questions and to improve the discussion of specific issues. Furthermore, we found a drop out of 70% in completing the questionnaire at T1. This could have been due to several factors. However, the most relevant aspect, in our opinion, was the delay in presenting the project to the teachers with respect to the school times. This determined either the withdrawal of a number of teachers that felt unable to complete the program, or the difficulty in administering the questionnaire at T1 because the end of the school year was approaching. The work herein presented represents the first step of an on-going activity to increase Food Literacy, and lay the foundation for a healthier relationship with food and the daily diet. In particular, the progressive journey towards a better knowledge of the vegetable world might reduce children’s reluctance to eat vegetables and favour awareness regarding the importance of consuming a balanced diet rich in fruits and vegetables. Such knowledge can provide children with a scientific basis to understand how to combine food in daily and weekly menus to maintain the right balance among nutrients and reinforce the concept that consuming a healthy and varied diet is the main tool to preserve human health. In conclusion, our study, although developed for Italian students, can be suitable for application in other countries, as it deals with public health issues and nutrition principles and recommendations widely accepted and shared. Moreover, the program is not limited to specific traditional or geographical contexts and can, thus, be implemented as an effective preventive action on public health.

## Figures and Tables

**Figure 1 nutrients-13-01547-f001:**
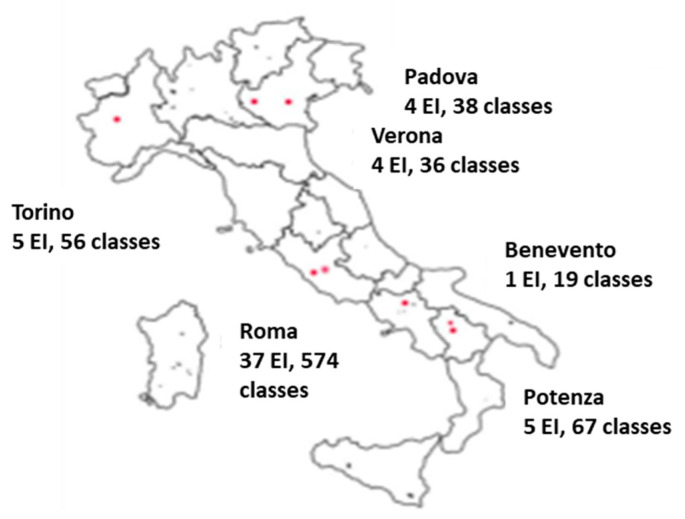
Geographic localization of the Educational Institutes (EI) and classes participating in the implementation of MaestraNatura program.

**Figure 2 nutrients-13-01547-f002:**
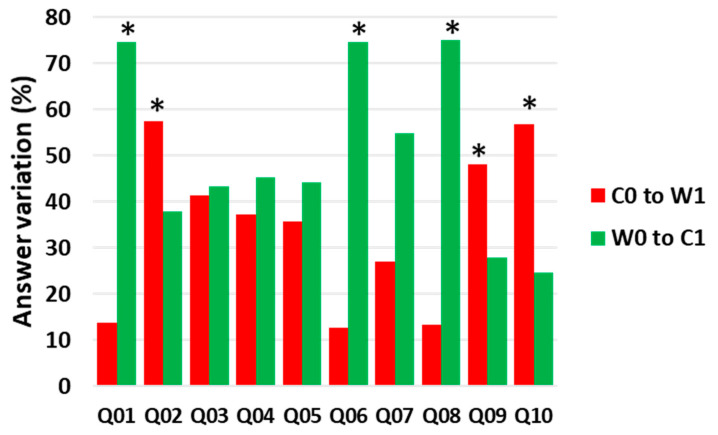
Answer variation to the questionnaires filled in at the end of the didactic path. Data represent the answer variation (%) of children that move from wrong to correct answer (W0 to C1) computed as [(W0 to C1)/W0] and from correct to wrong answer (C0 toW1) computed as [(C0 to W1)/C0]. C0, C1: correct answer at T0 or T1, respectively; W0, W1: wrong answer at T0 or T1, respectively. Q01–Q10: questions showed in Table 1, Panel B. Significance level *p* refers to Fisher’s exact probability test; * *p* < 0.05 when applying Bonferroni’s correction.

**Table 1 nutrients-13-01547-t001:** Questionnaire for the students.

Panel A: What do you know about food?
Question
What is the function of fats in our body? What is the function of sugar in our body? What is the function of water in our body? What is the function of proteins in our body? What is the function of vitamins in our body? What is in milk? What is in a steak? Which of the following foods provide more energy? Which of the following foods is a fruit? Which of the following foods is a vegetable? Which is the energy source for plants?
Panel B: What do you know about vegetables?
Question
Q01 Tomatoes are…
Q02 Fennel is…
Q03 Onions are…
Q04 Courgettes are…
Q05 Aubergines are…
Q06 Carrots are…
Q07 Stems develop into…
Q08 Roots allow the plant to…
Q09 Flowers are useful to…
Q10 What do seeds need to develop?

**Table 2 nutrients-13-01547-t002:** Didactic contents, experimental and practical activities, and objectives of the MN educational paths.

School Class Educational Path (hours)	Didactic Contents	Experimental Activities	Activities at Home	Objectives
I primary “The miracle of life” (8 h)	“The miracle of life”	Look at the bean germination Combine the card showing flower, fruit, leaves, seed, with the correct food Recognize the plant by touching or smelling it	Let’s prepare: Carrot oil, Vanilla extract, Cinnamon and apple cake, Sweetness with rose petals, Mint syrup	Encouraging the manipulation and transformation of food; discovering new flavours and food
II primary “Microorganisms, friends or enemies?” (8 h)	“Organic or Inorganic?” “Food storage methods”	Let’s breed yeast Let’s turn must into wine Let’s make vinegar and yogurt Let’s observe milk curdling	Let’s prepare: Bread with brewer’s yeast Home-made sourdough Bread with sourdough	Encouraging the handling and transformation of food; refining the sensitivity towards “genuine flavours” in children and families; introducing elements of food hygiene.
III primary “Water’s superpowers” (8 h)	“Water’s superpowers” “Leftovers and food waste”	Which substances dissolve in water? Experiments on solubility Experiments on surface tension and capillarity How do clouds form? Experiments on the water cycle	How does water freeze? Let’s prepare icicles Anti-waste recipes: reuse of leftovers Homemade cosmetics and detergents	Introducing children to the knowledge of water, an essential element for life; sensitizing children and families on food waste issues
IV primary “It’s too easy to say vegetables” (10 h)	“Seeds and fruits” “Food chains”	Seeds germination and dissemination Draw a poster classifying vegetables based on: (a) edible parts; (b) seasonality; (c) family Vegetable slicing and creation of a vegetable garden from kitchen waste Extract chlorophyll from leaves	Compile the identity card of seeds, underground drums, roots, leaves, fruit from the garden Let’s cook seasonal vegetables: educating taste	Recognizing the different parts of the plant and their function; learning which part of the most common vegetables we eat; discovering the variety of seasonal vegetables; developing awareness of the importance of consuming a varied diet, rich in fruit and vegetables to preserve health; reducing children’s unwillingness to eat vegetables.
V primary “Why do we have to eat?” (10 h)	“Why do we have to eat?” “Discover the egg” “Discover milk”	What’s in the egg? Is this egg fresh? What’s in milk? What food group does it belong to? Plan a daily menu	Let’s go cooking: with and without eggs with and without milk	Understanding the importance of food in maintaining human well-being; Understanding the origin of foods and how technology affects their availability. Distinguishing between food and nutrients; learning how to classify foods in food groups. Learning how to combine foods to plan a balanced daily menu
I secondary “Mindful eating: don’t de-vour the planet” (12 h)	“Food waste” "Food sustainability" “Food storage meth-ods” “Milk: from the sta-ble to the table”	Preparing milk products: Let’s make yoghurt Let’s make curd Let’s make butter Contrasting food waste: how does mould form? Read food labels properly How should we store food? Put the food in the fridge	Let’s go cooking: Brioche, sponge cake, breakfast cake, anti-waste recipes	Understanding what food waste is and what we can do to reduce it. Learning the meaning of environmental footprint and the importance of having a sustainable diet.
II secondary “We are what we eat” (14 h)	“The digestive process” “There is no perfect food” “Recognizing nutrients”	Simulating the digestion process Discover macronutrients providing energy What’s inside? How many times a day, how many times a week? Plan a weekly menu	Let’s go cooking: cream, crème caramel, meringues	Identifying human organs and their functions in digestive and metabolic processes; understanding the importance of healthy behaviours; identifying human organs and explaining their functions by means of models; drawing up a scientific report describing all the phases of an experiment; using tools and measurement units with confidence. Recognizing nutrients in food Learning the importance of a healthy, varied diet; learning how to combine foods in meals throughout the week to maintain the right variety of foods and the proper daily and weekly frequencies of consumption

**Table 3 nutrients-13-01547-t003:** Percentage of completing activities and teachers’ satisfaction for the proposed activities.

Activity	% of Completing Activities (Mean ± SD)	% of Teachers’ Positive Judgement (Mean ± SD)
Experimental laboratory	71 ± 20	90 ± 4
Activities at home	86 ± 5	94 ± 4
Training	96 ± 3	92 ± 3
Overall judgement on the project		100

**Table 4 nutrients-13-01547-t004:** Assessment of students’ basal knowledge on fruit and vegetables.

Question On	Total		3rd Class (*n* = 314)		4th Class (*n* = 276)		5th Class (*n* = 236)		*p*
	(*n* = 826)								
	*n*	%	*n*	%	*n*	%	*n*	%	
tomato	391	47.3	161	51.3	122	44.2	108	45.8	0.199
fennel	214	25.9	52	16.6	88	31.9	74	31.4	<0.001 *
courgette	260	31.5	105	33.4	82	29.7	73	30.9	0.606
onion	513	62.1	198	63.1	160	58.0	155	65.7	0.183
aubergine	271	32.8	109	34.7	86	31.2	76	32.2	0.636
carrot	579	70.1	210	66.9	195	70.7	174	73.7	0.214
stem	407	49.3	163	51.9	151	54.7	93	39.4	0.001 *
root	673	81.5	263	83.8	221	81.1	189	80.1	0.414
flower	457	55.3	153	48.7	158	57.3	146	61.9	0.007
seed	180	21.8	54	17.2	61	22.1	65	27.5	0.055

Percentages of correct answers to panel B questionnaire collected at the beginning of the educational activities (T0). Data are shown in total and by school class. Significance level *p* refers to Fisher’s exact probability test; * *p* < 0.05 when applying Bonferroni’s correction.

**Table 5 nutrients-13-01547-t005:** Evaluation of the knowledge increase at the end of didactical activities.

Question on	T0		T1		T1-T0	Wilcoxon
	(*n* = 246)		(*n* = 246)			*p*
	*n*	%	*n*	%	%	
tomato	116	47.2	197	80.1	32.9	
fennel	61	24.8	96	39.0	14.2	
courgette	75	30.5	118	48.0	17.5	
onion	142	57.7	136	55.3	−2.4	
aubergine	70	28.5	123	50.0	21.5	
carrot	159	64.6	204	82.9	18.3	
stem	133	54.1	159	64.6	10.6	
root	202	82.1	208	84.6	2.4	
flower	135	54.9	101	41.1	−13.8	
seed	51	20.7	70	28.5	7.7	
						0.032

Percentages of correct answers collected at the beginning (T0) and at the end (T1) of the educational activities. T1-T0 represents the percentage of knowledge increase for each question. Significance level *p* refers to Wilcoxon matched-pairs signed ranks test *p* < 0.05.

## Data Availability

The data presented in this study are available on request from the corresponding author.

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
