# Peer review of "Promoting Health and Food Literacy through Nutrition Education at Schools: The Italian Experience with MaestraNatura Program"

_nutrients, 2021, doi:10.3390/nu13051547_

Round 1

Reviewer 1 Report

It is an important research question, addressing one of the components to prevent NCDs which is knowledge. However, knowledge doesn't necessarily lead to change in behavior. 

I have few comments :

Title:

May be misleading, as the study did not examine change in behavior due to the nutrition education. Mat be to revise the title to " promoting Health and Food literacy......

Methods:

1- Need details on how 6 years old were interviewed. It is tricky to get answers from first graders. 

2- Looking at Table 1 page 4, some questions are hard for first graders (6 years old). Do authors asked all age groups from 6-11 years the same questions? Please explain

3- In the pilot study, need please to explain why they authors only recruited 3rd, 4th, and 6th classes.

Results:

Table 4 is confusing, need to look for a simpler way to present findings

Author Response

It is an important research question, addressing one of the components to prevent NCDs which is knowledge. However, knowledge doesn't necessarily lead to change in behavior. 

We thank the reviewer for the nice comment. We agree that knowledge does not necessarily lead to change in behaviour, but it may be a useful step to favour such change.

I have few comments:

Title:

May be misleading, as the study did not examine change in behavior due to the nutrition education. Mat be to revise the title to " promoting Health and Food literacy......

We agree with the suggestion and modified the title in: Promoting Health and Food Literacy through Nutrition Education at schools: the Italian experience with MaestraNatura program  

 Methods:

  • Need details on how 6 years old were interviewed. It is tricky to get answers from first graders. 
  • Looking at Table 1 page 4, some questions are hard for first graders (6 years old). Do authors asked all age groups from 6-11 years the same questions? Please explain

We thank the reviewer for prompting us to reconsider this point that was not explained in enough detail; we do apologize for this. The children aged 6-7 years were only administered the questions of the Table 1, panel B, and the questions were presented orally by the teachers.

  • In the pilot study, need please to explain why they authors only recruited 3rd, 4th, and 6th classes.

We chose these classes because the contents of the educational path to be tested were sufficiently in line with their curricular programs.

Results:

Table 4 is confusing, need to look for a simpler way to present findings

According with the reviewer comment, the results previously reported in Table 4 are now shown in two tables (newTable 4 and Table 5). The new Table 4 reports data on students’ basal knowledge and Table 5 reports the increase of knowledge evaluated by comparing the answers collected at T0 and T1.

Reviewer 2 Report

Overall, nutrition education programs to develop healthy eating behaviors is an area of critical importance globally, as early nutrition habits set patterns that persist throughout life.  The basis of this paper is strong, however, there are several areas of clarification needed throughout as mentioned below.

Abstract:

Line 21: Please specify who the approval rating was for – teachers?  Where is this presented in the results section?

Lines 21-22: In abstract and throughout the paper, it would be helpful to include age of children as terminology regarding school level can vary based on region around the world.  This was done in the results section (lines 170-171) but would also be helpful in other areas including the abstract.

Lines 25-26: “Results showed a significant increase in right answers at T1 with respect to T0 (z=2.017, p=0.044).”  This does not match the results section which states z=2.244, p=0.025.  Please clarify discrepancy.

Fishers Test results should be included in abstract– which shows mixed impact of the intervention.

Introduction:

Lines 72-75: What is a “traditional frontal lesson”?

Methods:

Lines 144-155: 2.3 Statistical analysis – Some of the information in these lines belongs in the results or discussion, not the methods section.

Results:

Lines 175-177: 3.1 Preliminary assessment of the main knowledge gaps in nutrition facts -- Please include the survey questions in an appendix.  The text refers to a specific question on the energy cycle, but the actual question is not provided.

Lines 190-193: 3.2 Definition of the nutrition education contents -- What was the source of information for the nutrition education?  For example, sustainable diets are still being defined.

Table 3: What does “% of commitment” mean?  Why is there a range?  What does “% satisfaction” mean?  Why is there a range? Why are both at 100 for overall judgement when it states the scale is 0-5?

3.3 Pilot study to test one of the educational paths -- This section is difficult to read and requires clarification.

Suggest breaking the single paragraph of 3.3 into different sections, based on the tables discussed. 

Students were given a multiple-choice questionnaire: Please clarify which questions. Table 1?

“…a similar variability in knowledge depending on the issue considered, the total knowledge appears to have slightly increased.”  Where are these numbers and statistical analysis?  Please clarify which numbers this statement is referring to.

“Specifically, for 7 questions out of 10 there was an increase in the right answer ranging from +10.6 to 32.9%, for 2 questions the percentage was quite similar to T0 and T1 (change equal to +2.4 and -2.4%), and for 1 out of 10 questions there was a decrease of -13.8%.” Please clarify where these numbers are coming from in Table 5.

The tables should have enough information to be able to stand alone.

Should Table 4 be Table 4a and Table 4b?

Did you consider presenting a table T0 for only those who completed T1 (n=246)?

30% completion at T1 is low – please provide more information regarding the low level of follow-up responses.

Table 5: Male vs. female is not needed – there is no difference.  Plain language would help make this table easier to read.  Details could be provided in the footnote.  For example, W0 à C1 could be “Knowledge Improved.”

Discussion:

Recommend including the survey questions in an appendix.

Discussion around the heterogeneity in the students’ responses depending on the issue in the preliminary survey is mentioned, however, this was also seen post intervention in the pilot.  Please include more information about these results (Fisher’s test).  Does this suggest there is still room for improvement in the program for certain topics?

Author Response

Overall, nutrition education programs to develop healthy eating behaviors is an area of critical importance globally, as early nutrition habits set patterns that persist throughout life.  The basis of this paper is strong, however, there are several areas of clarification needed throughout as mentioned below.

We thank the reviewer for the comments

Abstract:

Line 21: Please specify who the approval rating was for – teachers?  Where is this presented in the results section?

We modified the abstract to clarify that the approval was for teachers (line 22). This finding is shown in Table 3. We also modified the title of column to better explain what the data reported are related to.

Lines 21-22: In abstract and throughout the paper, it would be helpful to include age of children as terminology regarding school level can vary based on region around the world.  This was done in the results section (lines 170-171) but would also be helpful in other areas including the abstract.

Line 21-22: We thank the reviewer for this suggestion and added the information where it was lacking

Lines 25-26: “Results showed a significant increase in right answers at T1 with respect to T0 (z=2.017, p=0.044).”  This does not match the results section which states z=2.244, p=0.025.  Please clarify discrepancy.

We do apologize for the mistake, in both the text and the abstract previous statistics computed on different numbers were reported. We have now amended the abstract (lines 27-28) and the text (paragraph 3.3.2, line 275)

Fishers Test results should be included in abstract– which shows mixed impact of the intervention.

We thank the reviewer for this remark. We inserted in the abstract a sentence on Fisher’s exact probability test (lines 28-29). More details are reported in the Results section (section 3.3.2: lines 282-286 and Figure 2).

Introduction:

Lines 72-75: What is a “traditional frontal lesson”?

In ‘traditional frontal lesson’ the teacher stands at the front of the class and imparts his knowledge to the students. It belongs to the category of passive learning together with reading, listening, and watching movies, characterized by low percentage of knowledge retention (Dale E. Audiovisual methods in teaching. New York: Dryden Press; 1969.). This clarification is now in the introduction (Lines 87-88).

Methods:

Lines 144-155: 2.3 Statistical analysis – Some of the information in these lines belongs in the results or discussion, not the methods section.

The paragraph was entirely modified according to the reviewer’s suggestions.

Results:

Lines 175-177: 3.1 Preliminary assessment of the main knowledge gaps in nutrition facts -- Please include the survey questions in an appendix.  The text refers to a specific question on the energy cycle, but the actual question is not provided.

The survey questions were shown in Table 1 (panel A and B); sorry for not explaining adequately this point, that is now reported more clearly (line 177).

As regard the question on the energy cycle, we do apologize for the mistakes done. First of all, we asked two questions (8 and 11 in Table 1 Panel A) on energy issues and not on energy cycle. The second error we did, was not to report one of the two questions on this issue in Table 1, panel A (now reported as question number 11). Actually, quite often, starting from the third classes of primary school, the teachers talk about solar energy and how it is used by the plants. The text was modified accordingly (Lines 179 and 194).

Lines 190-193: 3.2 Definition of the nutrition education contents -- What was the source of information for the nutrition education?  For example, sustainable diets are still being defined.

The nutrition education contents were defined taking in account the WHO and national dietary guidelines as now reported in the text. As for the sustainable diet we referred to FAO guidelines and United Nations documents to draw from them basic principles on sustainability (lines 204-206).

Table 3: What does “% of commitment” mean?  Why is there a range?  What does “% satisfaction” mean?  Why is there a range? Why are both at 100 for overall judgement when it states the scale is 0-5?

We thank the reviewer for this comment that prompted us to better clarify these points. The commitment indicates the percentage of completing activities provided by the program. We modified the label of the correspondent column in Table 3.

The approval degree was evaluated by asking the teachers a positive or negative judgement on the different contents of the program. The ranges indicated the minimum and maximum percent of positive answers obtained; however, we changed the range into mean values + SD that result clearer.

Regarding the overall judgement, as now described in the Methods (Lines 127-130), the teachers were asked to evaluate the entire project with a score ranging from 0 to 5; the answers > 2 were considered positive. The percentage of teachers that gave a score >2 was 100%.  The 100% was erroneously reported also in the ‘commitment’ column and it was deleted.

3.3 Pilot study to test one of the educational paths -- This section is difficult to read and requires clarification. Suggest breaking the single paragraph of 3.3 into different sections, based on the tables discussed. Students were given a multiple-choice questionnaire: Please clarify which questions. Table 1?

We thank the reviewer for this comment. According to the reviewer’s suggestions the paragraph was divided in two sections based on the description of results reported in the tables. The questionnaire administered to the students for the pilot study was reported in Table 1, Panel B. This information is now reported in the text (line 257).

“…a similar variability in knowledge depending on the issue considered, the total knowledge appears to have slightly increased.”  Where are these numbers and statistical analysis?  Please clarify which numbers this statement is referring to.

“Specifically, for 7 questions out of 10 there was an increase in the right answer ranging from +10.6 to 32.9%, for 2 questions the percentage was quite similar to T0 and T1 (change equal to +2.4 and -2.4%), and for 1 out of 10 questions there was a decrease of -13.8%.” Please clarify where these numbers are coming from in Table 5.

We apologize for the lacking of clarity of this section. This paragraph was completely changed (lines 254-286). To better clarify the results we have also modified table 4 and added new table (Table 5) reporting data from subjects who filled in the questionnaire at T0 and T1.  In addition, the previous Table 5 was eliminated and those data are now reported in Figure 2.

The tables should have enough information to be able to stand alone.Should Table 4 be Table 4a and Table 4b?

The missing information was added and now the Tables should be able to stand alone. Table 4 was modified and divided in Table 4 and Table 5.

Did you consider presenting a table T0 for only those who completed T1 (n=246)?

As said above, we have added these data in the new Table 5

30% completion at T1 is low – please provide more information regarding the low level of follow-up responses.

We found a drop out of 70% in completing the questionnaire at T1. This could have been due to several factors. However, the most relevant aspect, in our opinion, was the delay in presenting the project to the teachers with respect to the school times. This led either to the withdrawal of a number of teachers that felt unable to complete the program, or, for other teachers, to the difficulty in administering the questionnaire at T1 because the end of the school year was approaching. This criticism is now reported in the discussion (lines 357-363). However, this is a clear demonstration of the usefulness of a pilot study that highlights critical points and allowing to modify the future experimental plan in order to overcome them.

Table 5: Male vs. female is not needed – there is no difference.  Plain language would help make this table easier to read.  Details could be provided in the footnote.  For example, W0 à C1 could be “Knowledge Improved.”

This comment was very useful for making these data easier to understand.  As suggested by the reviewer data on male vs female were eliminated. Moreover, in place of the table, we included a figure (Figure 2) that should result more understandable.

Discussion:

Recommend including the survey questions in an appendix.

All the questions are reported in Table 1, Panel A and B.

Discussion around the heterogeneity in the students’ responses depending on the issue in the preliminary survey is mentioned, however, this was also seen post intervention in the pilot.  Please include more information about these results (Fisher’s test).  Does this suggest there is still room for improvement in the program for certain topics?

As suggested by the reviewer more information about answer variability was included in the revised version (lines 354-357). The students failed more the answers to the questions about the plant functionality than on plant recognizing. These findings led us to rethink the wording of the questions and to improve the educational contents on these issues.

Reviewer 3 Report

This is a very interesting article which develops a comprehensive Food Education through Schools programme in Italy. The approach is appropriate and the sample is large enough to be able to successfully implement the programme throughout the country.
However, in my opinion, the introduction should be expanded and this interesting initiative should be contextualised in the European policies that since 2004 have been urging member countries to implement programmes on nutrition and physical activity to reverse the figures of childhood overweight and obesity.
The article should also indicate whether there are any socio-economic differences between students, as it is known that more disadvantaged groups have higher levels of overweight and obesity. Do such differences exist in the population? If so, what should be done to include concepts such as equity in the development of interventions? 
Regarding the questionnaires used, it is understood that they have been developed ad hoc for this work. Are there no validated questionnaires in Italy to assess knowledge in schoolchildren? If so, please justify.
It would also be interesting to include in the supplementary materials section of the journal a brief summary of each of the sessions so that the reader can get a better idea of the programme. The questionnaires used could also be included, since in order to understand the results, it is necessary to know the wording of the questions.

Author Response

This is a very interesting article which develops a comprehensive Food Education through Schools programme in Italy. The approach is appropriate and the sample is large enough to be able to successfully implement the programme throughout the country.

We are really grateful for the reviewer’s comment.

However, in my opinion, the introduction should be expanded and this interesting initiative should be contextualised in the European policies that since 2004 have been urging member countries to implement programmes on nutrition and physical activity to reverse the figures of childhood overweight and obesity.

We thank the reviewer for this suggestion. Some sentences on the European policy on obesity and overweight were added in Introduction (lines 67-80). The article should also indicate whether there are any socio-economic differences between students, as it is known that more disadvantaged groups have higher levels of overweight and obesity. Do such differences exist in the population? If so, what should be done to include concepts such as equity in the development of interventions? 

We agree with the reviewer that collecting data on the socio-economic status of students would provide very interesting information. It is well known that the prevalence of overweight and obesity is higher in people with low income and low educational level. Unfortunately, we did not collect data about these aspects in our pilot study. This decision was driven by different reasons, namely the desire not to overload too much the students with too many questions, and the specific objective of the pilot project that was to determine the effectiveness of the educational program in improving knowledge. The program, indeed, should contribute to overcoming the disadvantages due to socio-economic differences among students.

Regarding the questionnaires used, it is understood that they have been developed ad hoc for this work. Are there no validated questionnaires in Italy to assess knowledge in school children? If so, please justify.

To the best of our knowledge, do not exist validated questionnaires in Italy to assess knowledge in school children.

It would also be interesting to include in the supplementary materials section of the journal a brief summary of each of the sessions so that the reader can get a better idea of the programme. The questionnaires used could also be included, since in order to understand the results, it is necessary to know the wording of the questions.

The contents of the program are summarized in Table 2.  For each school class the contents of the sessions are shown, as well as the type of experimental/practical activities, and the objectives that are desired to reach. The questions asked in the preliminary survey and in the pilot study, are shown in Table 1, panel A and panel B.

Round 2

Reviewer 2 Report

The edits have helped to clarify several points throughout the paper. I have no additional comments.